# The stability of subducted glaucophane with the Earth's secular cooling

Yoonah Bang [1], Huijeong Hwang [1], Taehyun Kim [1], Hyunchae Cynn [2], Yong Park [3], Haemyeong Jung [3], Changyong Park [4], Dmitry Popov[4], Vitali B. Prakapenka [5], Lin Wang[6], Hanns-Peter Liermann [7], Tetsuo Irifune[8], Ho-Kwang Mao[6] & Yongjae Lee [1] ✉

The blueschist to eclogite transition is one of the major geochemical–metamorphic processes typifying the subduction zone, which releases fluids triggering earthquakes and arc volcanism. Although glaucophane is an index hydrous mineral for the blueschist facies, its stability at mantle depths in diverse subduction regimes of contemporary and early Earth has not been experimentally determined. Here, we show that the maximum depth of glaucophane stability increases with decreasing thermal gradients of the subduction system. Along cold subduction geotherm, glaucophane remains stable down ca. 240 km depth, whereas it dehydrates and breaks down at as shallow as ca. 40 km depth under warm subduction geotherm or the Proterozoic tectonic setting. Our results imply that secular cooling of the Earth has extended the stability of glaucophane and consequently enabled the transportation of water into deeper interior of the Earth, suppressing arc magmatism, volcanism, and seismic activities along subduction zones.

[1] Department of Earth System Sciences, Yonsei University, Seoul, South Korea. [2] Physics Division, Physical and Life Sciences Directorate, Lawrence Livermore National Laboratory, Livermore, CA, USA. [3] School of Earth and Environmental Sciences, Seoul National University, Seoul, South Korea. [4] High Pressure Collaborative Access Team, X-ray Science Division, Argonne National Laboratory, Argonne, IL, USA. [5] Center for Advanced Radiation Sources, University of Chicago, Argonne, IL, USA. [6] Center for High Pressure Science & Technology Advanced Research, Shanghai, China. [7] Photon Sciences, Deutsches Elektronen-Synchrotron (DESY), Hamburg, Germany. [8] Geodynamics Research Center, Ehime University, Matsuyama, Ehime, Japan. ✉email: yongjaelee@yonsei.ac.kr

Plate tectonics such as subduction has been operating since the early Earth[1–6], possibly since the Hadean or Eoarchean[7]. The Archean mantle temperature is estimated to be ca. 1500–1650 °C, which is higher than the present mantle temperatures of ca. 1350 ± 50 °C (refs. [8,9]). Thus, geothermal gradients in the Archean are similar or even higher than those of modern-day warm subduction zones (~8–12 °C km$^{-1}$)[2–4]. Under the P–T conditions of warm subduction, oceanic crust may release two-thirds of its water <2 GPa, equivalent to ~60 km depth, while still delivering ca. 2 wt.% $H_2O$ to the deeper Earth[10–12]. On the other hand, the Earth has undergone secular cooling from 2.5–3.0 Ga ago by as much as 50–100 °C Ga$^{-1}$ over the last 3 Ga, while 100–150 °C Ga$^{-1}$ in the present because the surface heat loss exceeded internal heating[9,13]. In turn, such a cooling in the average mantle temperature has affected the tectonic processes of the Earth by facilitating the modern-style subduction with low thermal gradient slabs (~5–8 °C km$^{-1}$)[13–16]. These cold subduction zones may allow the oceanic crust to lose only about one-third of its water by 2 GPa or ~60 km depth[10–12], hence transport more water into the deeper Earth. The water (or fluid) released from subducting slabs buoyantly rises in the overlying mantle wedge or crust to lower the solidus temperature by ca. 200–400 °C inducing partial melting and volcanism[17,18]. The $H_2O$ transport into the deep Earth is realized by subducting hydrous minerals, which exhibit a range of stability dictated by the P–T regime of the subduction system[10,11,19]. It is therefore essential to investigate the stability of hydrous minerals as a function of diverse geothermal gradients in the past and present tectonic settings to fully understand the evolution of deep water cycling, and related geochemical and geophysical activities.

The major fluid carriers in subducting oceanic crust are hydrous minerals, such as lawsonite, chlorite, and amphiboles, where $H_2O$ is contained in the form of molecules and/or structural hydroxyl[20]. Lawsonite ($CaAl_2Si_2O_7(OH)_2 \cdot H_2O$) contains as much as ca. 11.2 wt.% $H_2O$ in both molecular and hydroxyl forms, whereas structural hydroxyls account for ca. 10–13 wt.% $H_2O$ in chlorites (($Mg,Fe)_5Al_2Si_3O_{10}(OH)_8$). Amphiboles carry ca. 1–3 wt.% $H_2O$, much smaller than that of lawsonite or chlorite, but represent the greatest $H_2O$ sink because amphiboles may account for a large portion of the metamorphosed oceanic crust by as much as 20–60 wt.% for basaltic (MORB) compositions[21]. Glaucophane ($Na_2(Mg,Fe)_3Al_2Si_8O_{22}(OH)_2$) is a sodic amphibole (Supplementary Fig. 1 and Supplementary Table 1), diagnostic for the blueschist facies together with either lawsonite or epidote. The role of amphiboles in basaltic slabs for the generation of arc magma has been experimentally studied for decades to determine whether and how the dehydration of amphiboles provides water to the overlying mantle wedge[22]. Such dehydration reactions responsible for the blueschist to eclogite transition indeed release a considerable amount of $H_2O$, which triggers intermediate-depth earthquakes via the dehydration embrittlement[23–25] and induces partial melting in the overlying mantle wedge leading to arc magmatism[18,26]. In general, the blueschist to eclogite transition is characterized by a suite of dehydration reactions involving the breakdown of amphiboles into pyroxenes[27,28], and lawsonite into the garnet–kyanite–coesite assemblage[29] at elevated P–T conditions[27,30]. The so-called "absence of blueschist" is thus linked to the global dehydration and breakdown of the blueschist in the Precambrian plate tectonic settings, where warm subduction was a predominant process for recycling of $H_2O$ (ref. [31]).

In order to understand the present-day subduction process of oceanic crust and gain insights into the evolution of deep water cycle as a function of the Earth's secular cooling, we have investigated the stability of glaucophane under P–T conditions mimicking cold and warm subduction geotherms together with the high thermal gradients for the Proterozoic tectonic setting[3] (Fig. 1 and see "Methods" section in Supplementary information). Using the thermal models of global subduction system[19], our experimental P–T conditions followed the geotherms of the North Cascadia and South Chile subduction zones, and the Tonga and Kermadec subduction zones, representing warm and cold subduction systems, respectively. The Proterozoic thermal gradients of 25–50 °C km$^{-1}$ in $\delta T/\delta$ Depth were based on the recent compilation of P–T data estimated from 456 localities encompassing the Eoarchean to Cenozoic Eras[3]. We have used both resistive-heated and laser-heated (LH) diamond-anvil cell (DAC) techniques for in situ and ex situ high-pressure and high-temperature (HP–HT) synchrotron X-ray powder diffraction (XRD) experiments up to 7.8(3) GPa and 1390 ± 30 °C. We have also utilized a Paris-Edinburgh cell (PEC)[32] to perform in situ reversal experiments on a mixture of reactants containing glaucophane and a 1000-ton multi-anvil press to retrieve the dehydration products of glaucophane from 3 GPa and 950 ± 5 °C. A modified Griggs apparatus was also employed to extend the observation of glaucophane to a natural epidote blueschist rock up to 2 GPa and 730 ± 10 °C condition. In order to ensure that our experiments represent equilibrium conditions, samples in DAC and large volume press runs were held at selected pressure and temperature conditions for 1 h or up to several hours. Here, we show that the dehydration of glaucophane strongly depends on the thermal gradients of the subduction zone, so that arc magmatism, volcanism, and seismic activities would have been suppressed by secular cooling and subsequent generation of cold subduction system, where water is transported deeper into the Earth.

## Results

**Stability of glaucophane along cold subduction zone.** In situ HP–HT XRD experiments on glaucophane were performed up to 7.8(3) GPa and 760 ± 45 °C for the slab surface, and up to 5.6(3) GPa and 450 ± 30 °C for the slab Moho under dry and wet with 4 wt.% of $H_2O$ conditions to follow the cold subduction geotherm of the Tonga and Kermadec thermal model[19] (Fig. 1, Supplementary Fig. 2, and Supplementary Table 2). Glaucophane under such cold subduction geotherm conditions remained stable up to 7.6(3) GPa and 660 ± 40 °C conditions, equivalent to ca. 240 km depth. This extends the stability of glaucophane compared to 3.1 (1) GPa at 700 ± 10 °C or 2.5(1) GPa at 840 ± 10 °C, as estimated in previous studies[33–35]. Our result is in agreement with the estimation that oceanic crust within cold subduction zone holds more water by ca. 2 wt.% than in warm subduction zone[10–12]. Subsequently, at higher pressure and temperature conditions of 7.8(3) GPa and 760 ± 45 °C, equivalent to ca. 245 km depth, we found that glaucophane dehydrates and breakdowns into pyroxenes and coesite as described below (Fig. 2 and Supplementary Fig. 5):

$$Na_2(Mg, Fe)_3Al_2Si_8O_{22}(OH)_2 = 2\ NaAlSi_2O_6 + 1.5(Mg_{1-x}Fe_x)_2Si_2O_6 + SiO_2 + H_2O$$

Glaucophane    Jadeite    Enstatite    Coesite    Fluid

(1)

In order to confirm the enhanced stability and provide a link to interpret seismic low-velocity layer along cold subduction zones, we determined the bulk modulus and linear compressibility of glaucophane at ambient and high temperature at 620 °C under diverse pressure media (Supplementary Fig. 4). Our derived bulk moduli are the same for the different pressure media within $2\sigma$ and agree well with the data reported in previous studies[36,37]. We identify that under high-temperature conditions, the anisotropy in linear compressibility is significantly modulated, i.e., at ambient temperature, $\beta_a = 5.1$ (kbar$^{-1} \times 10^{-4}$), while $\beta_b = 2.3$

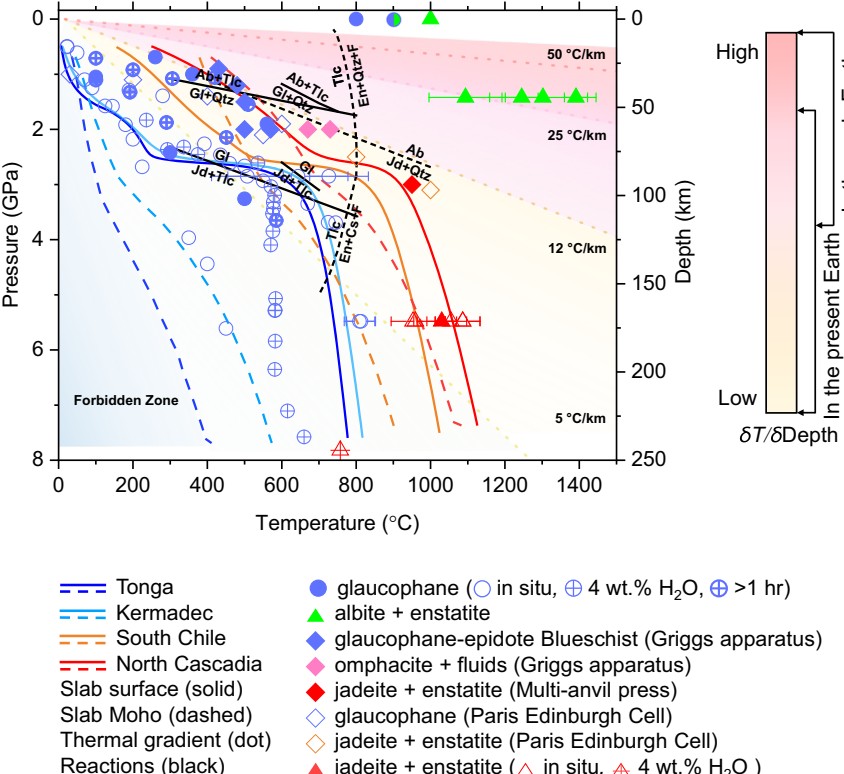

**Fig. 1 Stability of glaucophane at P–T conditions of different subduction geotherms.** Curved lines represent individual subduction geotherms from Syracuse et al.[19]. Continuous and dashed curves denote the P–T paths of subducting slab surfaces and corresponding slab Moho, respectively. The Tonga and Kermadec represent cold subducting slab, whereas the North Cascadia and South Chile represent warm subducting slabs. Black lines represent the upper- and lower-pressure stability of glaucophane from previous studies[33,34,39], while black dashed lines represent related reactions[40,61–63]. The colored trilateral regions from top to bottom represent high, intermediate, and low geothermal gradients ($\delta T/\delta$Depth), as defined by Brown and Johnson[3]. The forbidden zone is a P–T region of ultrahigh pressures, where many numerical models predict slab-top geotherms of <5 °C km$^{-1}$ (ref. [64]). High-temperature experiments at ambient pressure were performed as a reference (Supplementary Fig. 9). Phase abbreviations: glaucophane (Gl), jadeite (Jd), enstatite (En), albite (Ab), talc (Tlc), quartz (Qtz), coesite (Cs), and fluid (F).

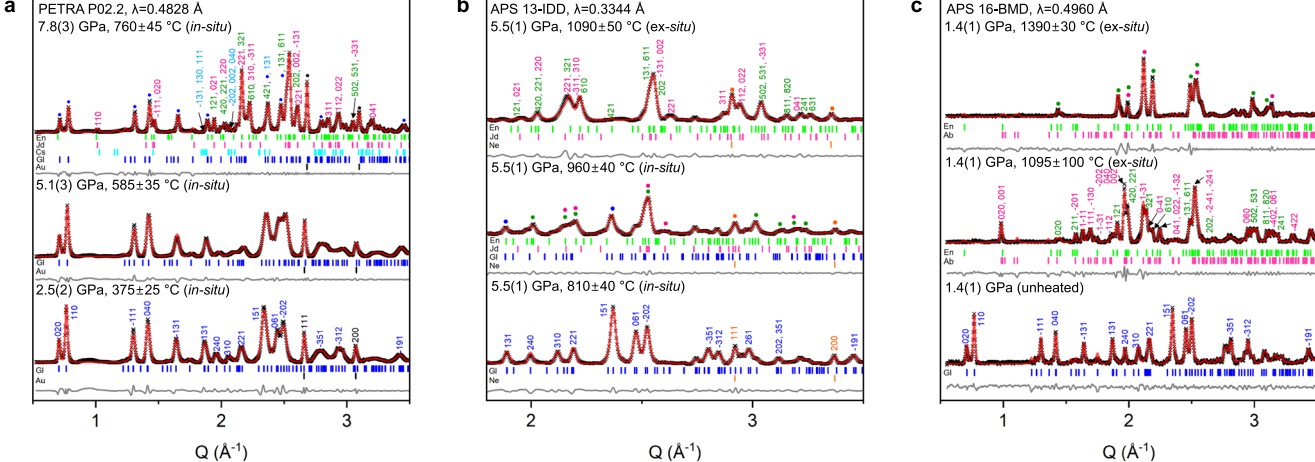

**Fig. 2 XRD patterns of glaucophane at P–T conditions of three different subduction geotherms.** Representative X-ray powder diffraction patterns of glaucophane along the P–T conditions of **a** cold subduction zone, **b** warm subduction zone, and **c** high thermal gradients in the Proterozoic tectonic settings. The experimental data and profile fits using the LeBail method[65,66] are shown in black symbols and red lines, respectively, with difference curves in gray lines. The backgrounds of the X-ray diffraction patterns have been subtracted prior to the data being fitted (see Supplementary Fig. 3 for the original patterns). Phase abbreviations: glaucophane (Gl), enstatite (En), jadeite (Jd), albite (Ab), coesite (Cs), gold (Au), and neon (Ne).

and $\beta_c = 2.2$ (kbar$^{-1} \times 10^{-4}$), whereas at high temperature, the $a$-axis compressibility is reduced to $\beta_a = 3.9$ (kbar$^{-1} \times 10^{-4}$), while $b$- and $c$-axes compressibilities are maintained to $\beta_b = 2.3$ and $\beta_c = 2.4$ (kbar$^{-1} \times 10^{-4}$), respectively.

**Dehydration and breakdown of glaucophane along warm subduction zone.** Ex situ and in situ HP–HT XRD experiments on glaucophane were performed up to 5.5(1) GPa and 1090 ± 50 °C for the slab surface and up to 5.5(1) GPa and 810 ± 40 °C for the slab Moho to follow the warm subduction geotherms of the North Cascadia and South Chile thermal models by Syracuse et al.[19] (Fig. 1 and Supplementary Table 2). The XRD data of the quenched samples after heating at the slab surface conditions above ca. 5.5(1) GPa and 1090 ± 50 °C, equivalent to ca. 170 km depth, were indexed to identify the breakdown products of glaucophane as in Eq. (1) (Fig. 2 and Supplementary Fig. 5). The same dehydration scheme, though different in depths, under cold and warm subduction conditions agrees well with the thermodynamic calculation[38]. Structural hydroxyls of glaucophane are released as fluid when it decomposes into jadeite-bearing assemblages of the eclogite facies. Our experimental results thus demonstrate that blueschist to eclogite transition can be simulated by the dehydration breakdown of glaucophane at different depths depending on the subduction geotherms.

In order to complement our experimental results using a single mineral phase in a DAC, we have further investigated the dehydration of glaucophane on a macroscopic scale using a natural epidote blueschist rock containing ca. 55 vol.% glaucophane. A 3 mm diameter core-drilled sample of blueschist rocks were heated up to 730 ± 10 °C at 2 GPa for 9 h, using a modified Griggs apparatus (Fig. 1 and Supplementary Table 2). The recovered sample showed that glaucophane has partially been dehydrated, and a new dehydration product, omphacite pyroxene (($Ca,Na$)($Mg,Fe,Al$)$Si_2O_6$), has formed, leaving trails of fluid inclusions in glaucophane crystals (Fig. 3 and Supplementary Fig. 7). Energy-dispersive spectroscopy confirmed the compositions of the recovered glaucophane and the new dehydration product omphacite, as shown in Supplementary Fig. 7. This result establishes that the dehydration of glaucophane in a natural blueschist rock begins near 670 ± 10 °C at 2 GPa conditions, which corresponds to ca. 60 km depth along warm subduction zone.

Furthermore, we have carried out reversal experiments along the P–T conditions of the warm subduction zones using a mixture of reactant minerals, i.e., glaucophane, jadeite, and talc, in the PEC. The mixture was heated above the breakdown condition of glaucophane up to 1000 ± 100 °C at 3.1(3) GPa over 4 h, and then cooled down to 550 ± 100 °C at 2.1(3) GPa over 6 h. At these P–T conditions, we observed the regrowth of glaucophane (130) peak, attesting the reaction boundary would be between ca. 50 and 100 km depths along the warm subduction geotherms (Supplementary Fig. 8).

**Subducting glaucophane in the high thermal gradients of the early tectonic setting.** Additional LH-DAC experiments were conducted to mimic the high thermal gradients model in the Proterozoic tectonic setting. At conditions of 1.4(1) GPa after heating between 1095 ± 100 and 1390 ± 30 °C, corresponding to a depth around ~40 km, glaucophane breaks down to albite and enstatite, and releases fluid $H_2O$ (ref. [39]; Fig. 2 and Supplementary Fig. 5), giving rise to the breakdown scheme:

$$Na_2(Mg,Fe)_3Al_2Si_8O_{22}(OH)_2 \rightarrow NaAlSi_2O_6 + (Mg_{1-x}Fe_x)_2Si_2O_6 + H_2O$$

| | | | |
|---|---|---|---|
| Glaucophane | Albite | Enstatite | Fluid |

(2)

This scheme is different from the Eq. (1) observed for the cold and warm subduction zones, but in the P–T relationship of the reaction, albite = jadeite + quartz, albite is known to be stable at lower pressures than jadeite[40]. The onset depth of dehydration breakdown of glaucophane thus appears to be inversely proportional to the thermal gradients ($\delta T/\delta Depth$) of the subduction system. We, however, note that the established depths for the dehydration could change when reaction rates are considered, which is beyond our current experimental capability. Johnson and Fegley reported that partial dehydration can be initiated in amphibole tremolite over several months at temperatures between 750 and 965 °C (refs. [41,42]).

## Discussion

At the same depth, temperature difference between cold and warm subduction zones ranges from ca. 175 °C to ca. 400 °C. According to the P–T conditions studied here (up to 7.8(3) GPa and 760 ± 45 °C), glaucophane would persist to depths of ca. 240 km in cold subduction zones with a geothermal gradient of ~5–8 °C km$^{-1}$. Along such cold subduction zones, the fully hydrated oceanic crust with the initial $H_2O$ content of ca. 6 wt.% may lose ca. 2 wt.% $H_2O$ by 2 GPa or ~60 km depth, while the rest would be transported deeper into the Earth[10–12]. This is in line

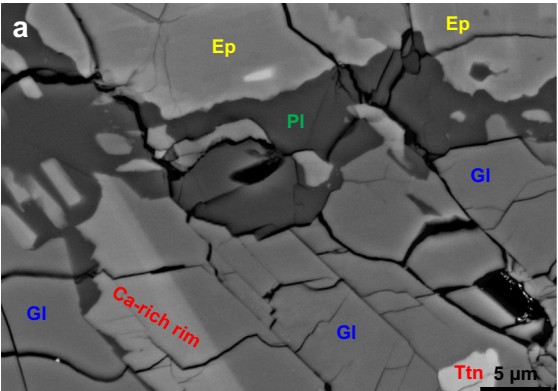
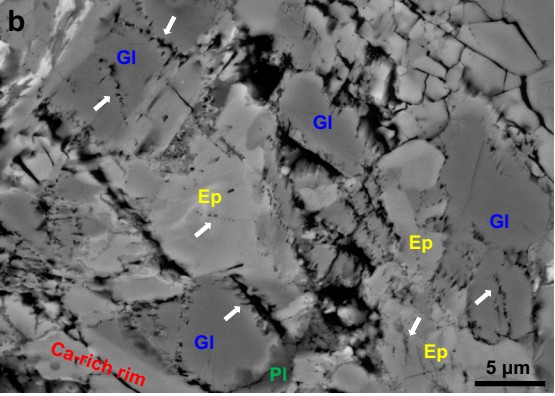

**Fig. 3 Experimental product of natural epidote blueschist showing dehydration features of glaucophane.** A modified Griggs apparatus was used for the dehydration of the natural epidote blueschist. Back-scattered electron (BSE) images showing **a** no dehydration of epidote blueschist after experiment at 2 GPa and 570 ± 10 °C and **b** partially dehydrated glaucophane (Gl) with rugged grain boundaries and fluid inclusion trails (white arrows) after experiment at 2 GPa and 730 ± 10 °C; consequently, omphacite (om) has formed as the dehydration product (Supplementary Fig. 7). Phase abbreviations: epidote (Ep), plagioclase (Pl), and titanite (Ttn).

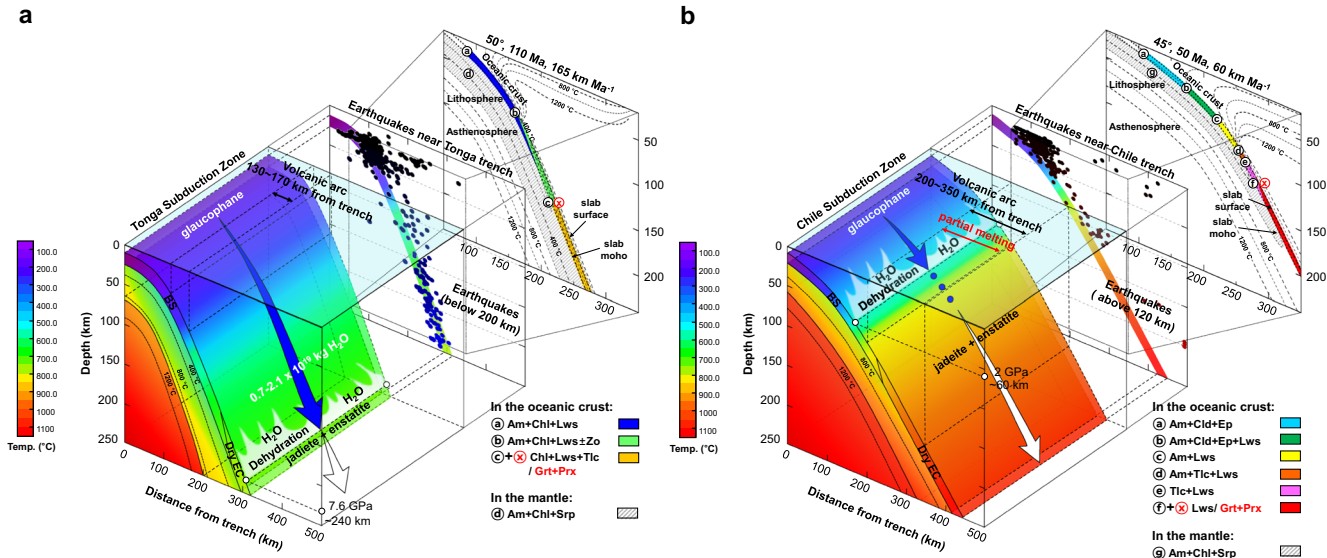

**Fig. 4 Subduction models of glaucophane (amphibole) in the contemporary subduction settings.** Subducting slab models representing **a** the Tonga (cold slabs) and **b** the Chile (warm slabs). The geometry and temperature profiles were adopted using the information available from the International Seismological Center (ISC-EHB)[67–69] and the thermal models by Syracuse et al.[19]. Glaucophane remains stable along the cold subduction zone down to ca. 240 km depth (blue arrow), whereas it dehydrates and decomposes into jadeite and enstatite along the warm subduction zone between 50 and 100 km depths (white arrow). Based on experiments on a natural blueschist rock, the onset depth of dehydration breakdown is estimated to be ~60 km depth. In the middle 2D projection layer, the earthquake frequencies are shown as filled circles using the data from the ISC-EHB. In the upper 2D projection layer, subducting hydrous minerals from Hacker et al.[70] and Magni et al.[71] are shown in colors on each thermal model (slab dip, slab age, and convergence rate of 50°, 110 Ma, 165 km Ma$^{-1}$ for the cold slab and 45°, 50 Ma, 60 °C km$^{-1}$ for the warm slab). Phase abbreviations: blueschist (BS), eclogite (EC), amphibole (Am), chlorite (Chl), chloritoid (Cld), lawsonite (Lws), epidote (Ep), zoisite (Zo), talc (Tlc), serpentine (Srp), garnet (Grt), and pyroxene (Prx).

with our observed stability of glaucophane under cold subduction conditions. On the other hand, glaucophane decomposes into pyroxenes. i.e., transition to eclogite, at shallower depths between 50 and 100 km in warm subduction zones with a geothermal gradient of ~8–12 °C km$^{-1}$. Upon dehydration, hydroxyls of glaucophane are released to form aqueous fluid, which would migrate upward to induce partial melting of the overlying mantle wedge or cause the lowering of solidus temperature in the subducting slab itself[43,44]. We estimate the average H$_2$O contents of glaucophane (and amphiboles) in the global oceanic crust to be in the range of 1.1–3.5 × 10$^4$ g H$_2$O m$^{-3}$ or 0.39–1.22 wt.% H$_2$O, which accounts for 7–20% of the total water content in the hydrated oceanic crust with overall 5–6 wt.% H$_2$O (Supplementary Table 4); such an amount, when released via dehydration reactions, would be sufficient to induce mantle melting and arc magmatism. In Fig. 4, we present the models of glaucophane stability together with the observed seismic frequencies and established mineral assemblages in two contrasting geothermal gradient settings. In cold subduction zones, glaucophane remains stable and enables water transport to deeper mantle. The amount of H$_2$O transported by glaucophane in global cold subduction zones is estimated to be as much as ca. 0.7–2.1 × 10$^{19}$ kg, which is approximately the amount of water in the Arctic ocean (Supplementary Table 4).

Our observation bears some implications for the distribution of seismic low-velocity layers, as well as seismic activities along the subducting slabs. According to our compressibility data (Supplementary Fig. 4), glaucophane behaves anisotropic even at high temperature, indicating strong mechanical resistance along the (100) plane, while [100] direction is relatively weak. Hydrous minerals have been suggested to be related to the seismic anisotropy and delayed seismic travel times along subduction zones in the depth range of 100–250 km (refs. [45–52]). The observed anisotropy and stability of glaucophane could, therefore, account for such seismic anomalies distributed, which would be deeper in the

colder and older slabs than in the warmer and younger slabs[53]. Furthermore, seismic observations reveal that the low-velocity layers spatially coincide with the zones of intermediate-depth earthquakes[54], which is in turn related to the dehydration of hydrous minerals[45,55]. With this regard, we show the correlation between the seismic frequencies along subduction zones and the stability range of glaucophane, i.e., the maximum depth of intraslab earthquakes ranges between 50 and 70 km in warm subduction zones, whereas it extends down to over 200 km in cold subduction zones[56] (Fig. 4 and Supplementary Fig. 11). As dehydration embrittlement of serpentine was previously proposed as a possible mechanism for the intermediate-depth earthquakes[23], subduction geotherm-dependent breakdown of glaucophane would provide another venue to explain the distribution of intermediate-depth earthquakes[57–59]. Our results would therefore serve as an experimental evidence to support the recent observation that the double seismic zone in the Tonga subduction system extends to deeper depths down to ca. 300 km (ref. [60]).

Among the 56 subduction geotherm data from Syracuse et al.[19], we could categorize 16 subduction zones as cold subduction system on the basis of the thermal parameter with average values of 48.9 °, 119.7 Ma, 74.8 km Ma$^{-1}$ for slab dip, age, and convergence rate, respectively (Supplementary Table 5 and Supplementary Fig. 10). Inferred from our results, sodic amphiboles, e.g., glaucophane, would be stable to deeper depths in ca. 28.5% of the global subduction system in the present Earth. Recent studies indicate that plate tectonics based on subduction-related processes has been initiated during the Proterozoic eon[2–6] or as early as 3.8 or 4.4 Ga (refs. [1,7]), when the majority of subduction zones would be categorized as warm or intermediate-to-high thermal gradients system[3]. According to our experimental results, the dehydration depth of glaucophane has increased with decreasing thermal gradients hence with secular cooling, which would translate to transportation of water into deeper Earth (Supplementary Table 4). We, therefore, conjecture that arc

magmatism would have been more effective via ubiquitous transformation of blueschist to eclogite in the high thermal gradients system of the early Earth. As the Earth undergoes secular cooling, progressively colder subduction zones have emerged and resulted in ca. 28.5% of the whole subduction system in the present Earth. Consequently, arc magmatism, volcanism, and related seismic activities linked to the dehydration of amphiboles have been globally suppressed, enabling blueschist to persist and be preserved in today's geodynamic system. The "absence of blueschist" from the Precambrian rocks might thus be explained by the subduction geotherm-dependent dehydration of glaucophane. On the other hand, subduction efficiency, i.e., the proportional amount of subducted $H_2O$ passing through the subduction zone filter[11], would have increased with the generation of low thermal gradients system toward the present Earth, as observed in our study.

## Data availability
All data generated or analyzed during this study are included with this published article and its Supplementary Information.

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

## Acknowledgements

This work was supported by the Leader Researcher program (NRF-2018R1A3B1052042) of the Korean Ministry of Science and ICT (MSIT). We also thank the partial supports by NRF-2016K1A4A3914691, NRF-2019K1A3A7A09033395, and NRF-2020R1A2C2003765 grants of the MSIT. Synchrotron experiments were performed at the beamlines 3D and 5A at PLS-II, HPCAT and GSECARS at APS, and ECB P02.2 at PETRA-III. HPCAT operations are supported by DOE NNSA's Office of Experimental Sciences. GSECARS is supported by the NSF-Earth Sciences (EAR-1634415) and Department of Energy (DOE)-GeoSciences (DE-FG02-94ER14466). The Advanced Photon Source is a U.S. Department of Energy (DOE) Office of Science User Facility operated for the DOE Office of Science by Argonne National Laboratory under Contract No. DE-AC02-06CH11357. We acknowledge DESY (Hamburg, Germany), a member of the Helmholtz Association HGF, for the provision of experimental facilities. H.C. thanks the support by the U.S. Department of Energy by the Lawrence Livermore National Laboratory under Contract No. DE-AC52-07NA27344. The authors thank Xueyan Du and Yuyong Xiong at HPSTAR for assisting laser-heating experiment, Youmo Zhou and Toru Shinmei at Ehime University for multi-anvil press experiment, Guoyin Shen, Rostislav Hrubiak and Curtis Kenney-Benson at HPCAT for supporting Paris-Edinburgh Cell experiment, G. Diego Gatta at the University of Milan for providing a natural sample of jadeite, and Moonsup Cho at Chungbuk National University and Sang-Heon Dan Shim at Arizona State University for valuable discussions.

## Author contributions

Y.B. contributed to the experiments and data analysis with the help from H.H, T.K., Y.P., C.P., D.P., V.P., and H.-P.L. Y.L. designed the research, discussed the results with H.C., H.J., L.W., T.I., and H.-K.M. and worked on the manuscript with all authors.

## Competing interests

The authors declare no competing interests.
