## [Peer Review File · Nature Communications]

REVIEWER COMMENTS

Reviewer #1 (Remarks to the Author):

This study by Bang and co-authors brings new experimental data on glaucophane stability at HP-LT conditions and discusses the implications for subduction processes and related seismicity and volcanism.

Being not an experimentalist myself I did not concentrate my comments on the quality and validity of the newly acquired data. I assume that the experimental results are correct.

Although the implications of this study are important, the paper lacks structure and the overall manuscript is rather unfocused. While the overall message is that glaucophane is more stable in colder environment, the implications for modern day and Archean conditions are somehow mixed-up and not appropriately supported/discussed.

Moreover, some figures or additional data must be provided. For instance the authors discuss the fact that their results are in agreement with seismic data. However they do not provide any data compilation or figure to support their statement.

Overall I think that the manuscript would greatly benefit from a more multidisciplinary approach, especially to support their current discussion. In my opinion their experiments on glaucophane stability have a broad impact in the geoscience community and their manuscript should reflect this better.

For instance the authors mainly discuss the results of their experiments in the light of "cold" vs "warm" subduction zone. This would greatly benefit from having "simple" 2D thermo-mechanical models with imposed velocities. Using Obtained P-T conditions from the models could help the authors discuss their results (on top of better presenting them) by quantifying the water fluxes based on their experiments and other data available in the literature (including petrogenetic grids). In a similar manner this would also allow them to explore various potential mantle temperatures and hence Archean conditions and the role of secular cooling.

Adding a simple mechanical relationship between the remaining water content stored in the oceanic crust and strength of the interface would also help to better understand the conditions allowing for the establishment of modern-day like plate tectonics.

At this stage I feel like the paper is not in shape to be accepted in a Journal such as Nature Communication. However, the core data and results from this study are relevant and potentially of broad impact.

In this light, I recommend a rejection of the manuscript with a invitation to resubmit once the implications of the experimental results are better quantified/discussed

Nicolas Riel

Reviewer #2 (Remarks to the Author):

Review of 'The stability of subducted glaucophane with the Earth's secular cooling.'

This manuscript describes a series of high-pressure and high temperature experiments that seeks to chart the stability of the amphibole mineral glaucophane under the variety of thermal gradients of the Earth's crust. The authors have used a number of different experimental approaches to show that the glaucophane decomposing within the Earth's crust to albite and enstatite would occur areas of warm subduction, but not areas of cold subduction. The authors then point out the impact of this finding on release of water into the earth's subsurface over time.

Though the impact of the results is well reasoned, I have found it challenging to see how the experimental results have been reasoned. Much of this is because of the large range of experimental methods that are used and that the supplementary information does not give all the information require to ratify the results.

My chief concern with the manuscript itself is the presentation of the diffraction results, in Figure 2 and in the supplementary files, which though I can see demonstrate the general point that they authors claim, has not been presented to the rigorous standard that I would have expected. There are a number of concerns I wish to raise:

- There is no presentation of the model fit to the data – only tick marks indicating the expected presence of Bragg peaks.
- I also assume the data have been reduced from their original format – especially the DAC measurements I would image that a background has been subtracted prior to the data being plotted in figure 2? Details of this should be given in the paper and it may help explain the differences in background between the PETRA and APS data.
- As the results of the paper depend on a series of results collected a different wavelengths it would be extremely beneficial for all the diffraction data to be plotted on a non-wavelength dependant x-axis (d-spacing or q space), currently it is really impossible (especially as the reduced data has only been made available in the figure) to compare each of them.
- By presenting the fits then the authors could also extract the induvial scattering from each phase that contributes to the fit, and also explain why peaks that could be from a number of phases had been attributed to one (for instance the co-occurrence of jadite/enstatite peaks in Figure 4 a. I very much dislike the 'labelling' of the peaks that has been done, because it is not consistent (not all peaks are labelled) and not very informative – it could even be misleading.
- Is there a reason that a Reitveld refinement hasn't been undertaken on the single phase patterns of glaucophane at LT/LP conditions? Doing this for at least one of the patterns (say the unheated pattern from the PE press) would show that there were not impurities and demonstrate the starting point of the experiments much better.

Additional points that I would like to see addressed:

- There are no uncertainties quoted for the temperature measurements. This is especially important as the manuscript presents data where high temperatures were generated in a number of different ways. Though in some cases (the resistive heating for instance) an uncertainty in the measurement is described in the supporting information – this isn't the case for the sample subjected to laser heating. I would like to see an uncertainty quoted for every temperature measured in the manuscript.
- The same point can be applied to the pressure measurements too – again multiple pressure measurement methodology was used (ruby fluorescence, multi-anvil calibration and MgO diffraction) uncertainties of each measurement must be quoted in the manuscript for clarity.
- Line 97 please add a reference to the Paris-Edinburgh press development
- Line 131, what is the relation of the compressibility parameters β_a , β_b , and β_c ? No equation (or reference to one) is given and how have these been derived from the collected data?
- Line 158 I wonder why that diffraction wasn't pursued for analysis of these samples?
- Line 204 I'm concerned about the statement 'possible down to the upper mantle-transition zone (~410km)' – how is this justified by the experimental program as the pressures that the

experiments went to were enough to replicate this depth?

- Line 223 amend text to 'modelled by Nair and Chacko44'
- Figure 1, what is the Forbidden zone? This label is not referred to in the figure caption or in the manuscript text
- Figure 4, There is no indication of expected temperatures on this figure, could this be done with a colour scale?

Seoul, September 18, 2020

Dear Reviewers,

We appreciate all of your comments and suggestions for our manuscript to Nature Communications (NCOMMS-20-23298) entitled: “The stability of subducted glaucophane with the Earth’s secular cooling” by *Yoonah Bang, Huijeong Hwang, Taehyun Kim, Hyunchoe Cynn, Yong Park, Haemyeong Jung, Changyong Park, Dmitry Popov, Vitali Prakapenka, Lin Wang, Hanns-Peter Liermann, Tetsuo Irifune, Ho-Kwang Mao and Yongjae Lee*. Please find attached our revised version of the paper and we have addressed in this revision all the reviewers’ comments and criticisms as described in this letter.

The three key issues the reviewers raised have been addressed as below:

1. Relationship between the stability of glaucophane and seismicity in the subduction zones (by reviewer #1): We have addressed this issue by comparing the stability of glaucophane to the earthquake frequencies in the representative cold (Tonga trench) and warm (Chile trench) subduction systems using the data from the International Seismological Centre (ISC) (revised Fig. 4 and new Supplementary Fig. 11). We confirm that the depth-dependent distribution of the earthquake occurrence bears strong relationship with the stability depth of glaucophane along the cold and warm subduction zones.

2. 2D thermo-mechanical model (by reviewer #1): As the editor pointed out, adding this multidisciplinary approach would be beneficial but beyond the scope of our current work. Instead, in this revision, we have added an alternative to conducting a 2D geodynamical modeling by incorporating the geometry and temperature profile data into the respective cold (Tonga trench) and warm (Chile trench) subduction systems using the information available from the ISC and the thermal models by Syracuse et al. (2010) (revised Fig. 4). We intend to seek and incorporate a 2D geodynamic modeling in our future work since we are in progress to extend the investigation on glaucophane into deeper depths.
3. Presentation of diffraction results (by reviewers #1 and #2): For all our diffraction data presented in this manuscript (main text and supplementary), we have performed the profile fits to confirm the phase assignments using the LeBail method (Le Bail et al. (1988), (2005)) and present them in q-axis to better compare the peak positions in a wavelength-independent scale (revised Fig. 2 and Supplementary Fig. 3, 6, 8, 9). We have also performed the Rietveld refinement (Larson & Von Dreele (1986) and Toby (2001)) using the diffraction data of glaucophane measured at ambient conditions and present the results to confirm its phase purity and structural characteristics (revised Supplementary Fig.1 and new Supplementary Table 1).

Full details of the reviewers' comments and our point-by-point responses are summarized below (all the changes made in the revised version are marked in red):

Reviewer #1 (Remarks to the Author)

Comments: This study by Bang and co-authors brings new experimental data on glaucophane stability at HP-LT conditions and discusses the implications for subduction processes and related seismicity and volcanism.

Being not an experimentalist myself I did not concentrate my comments on the quality and validity of the newly acquired data. I assume that the experimental results are correct.

Although the implications of this study are important, the paper lacks structure and the overall manuscript is rather unfocused. While the overall message is that glaucophane is more stable in colder environment, the implications for modern day and Archean conditions are somehow mixed-up and not appropriately supported/discussed.

Moreover, some figures or additional data must be provided. For instance, the authors discuss the fact that their results are in agreement with seismic data. However, they do not provide any data compilation or figure to support their statement.

Overall, I think that the manuscript would greatly benefit from a more multidisciplinary approach, especially to support their current discussion. In my opinion their experiments on glaucophane stability have a broad impact in the geoscience community and their manuscript should reflect this better.

Reply: We appreciate the overall positive comments and constructive criticisms by reviewer #1. We have addressed the issue on the seismic data as described in the key revision point #1 above. We have incorporated the earthquake occurrence data along the Tonga and Chile trenches over the past 10 years as compiled in the ISC Bulletin. As the result, we show that the earthquakes along the cold subduction zone are distributed over the depth of 200 km while they are focused below the depth of 60 km along the warm subduction zone, which is in close agreement with our observed stability ranges of glaucophane along the respective subduction environments (revised Fig. 4 and new Supplementary Fig. 11).

Comments: For instance, the authors mainly discuss the results of their experiments in the light of "cold" vs "warm" subduction zone. This would greatly benefit from having "simple" 2D thermo-mechanical models with imposed velocities. Using Obtained P-T conditions from the models could help the authors discuss their results (on top on better presenting them) by quantifying the water fluxes based on their experiments and other data available in the literature (including petrogenetic grids). In a similar manner this would also allow them to explore various potential mantle temperatures and hence Archean conditions and the role of secular cooling.

Adding a simple mechanical relationship between the remaining water content stored in

the oceanic crust and strength of the interface would also help to better understand the conditions allowing for the establishment of modern-day like plate tectonics. At this stage I feel like the paper is not in shape to be accepted in a Journal such as Nature Communication. However, the core data and results from this study are relevant and potentially of broad impact.

In this light, I recommend a rejection of the manuscript with a invitation to resubmit once the implications of the experimental results are better quantified/discussed

Reply: The issue on adding a 2D thermo-mechanical model is addressed in the key revision point #2 above. As an alternative to conducting such a 2D geodynamical modeling, we have incorporated the geometry and temperature profile information into the cold (Tonga trench) and warm (Chile trench) subduction systems to better distinguish respective geothermal environments using the data from the ISC and the thermal models by Syracuse et al. (2010). As stated above, it is our plan to seek for appropriate geodynamical modeling in our work in progress on glaucophane at deeper depths (revised Fig. 4).

Comments made on the manuscript: *“The Earth has undergone secular cooling from 2.5-3.0 Ga ago to the present day by as much as loosely constrained average value of 100-150 C/Ga over the last 3 Ga” That would mean that the temperature was 250-450 °C higher during archean, which is much higher than 150-250 °C.*

Reply: We have revised the cooling rate description as indicated below: “The Archean potential mantle temperature is estimated to be in the range of 1500-1650 °C, and the Earth is estimated to have undergone secular cooling at a rate of 50-100 °C/Ga over the past 3 Ga while 100-150 °C/Ga in the present”. (p2, line 43, 48-49)

Comments made on the manuscript: *“As a result, subduction zones with low thermal gradients such as the Tonga and Kermadec subduction zone have developed in the present global subduction system.” Cooling of the Earth led to warm and cold subduction zones, what is your point here? To me the difference in geothermal gradients is the result of multiple controlling factors.*

Reply: We have added new sentences with references to support the initiation of subduction driven plate tectonics as a function of the Earth’s secular cooling. Subduction zones with low thermal gradients can be explained as a result of the secular cooling that ultimately produced the present tectonic systems: In Weller and Lenardic (2018, reference #15), the authors suggest that the Earth’s internal heat budget fundamentally controls the initiation of subduction-driven tectonics. In Ganne and Feng (2017, reference #14), the progressive steepening of subduction zones throughout Proterozoic era is explained as the result of secular cooling. Over geological time, subduction systems with low thermal gradients such as the Tonga and Kermadec

subduction zones have developed into the present global subduction environments as the continued secular cooling of the mantle has controlled the initiation of subduction-driven plate tectonics and steepening of subducting slabs. (p2-3, line 50-53)

Comments made on the manuscript: *“The H₂O influx into the deeper depths, therefore, does not simply depend on the subduction system governed by different P-T regimes via subduction rate, angle, and age, etc., but should also be rebalanced by the secular cooling of the mantle itself, manifesting the need to consider the geothermal gradients in the past and present tectonic settings to fully understand the evolution of deep water recycling.” I don't quite understand why P-T conditions and subduction regime is not sufficient to describe deep water recycling. Do you want to say that Secular cooling plays a first order control in the P-T and subduction regime and therefore on fluid recycling in the mantle?*

Reply: We like to emphasize that both secular cooling of the mantle and multiple factors such as convergence rate, slab angle, and slab age would mutually/interactively control the subduction zones and the related water influx into the interior of the Earth.

Comments made on the manuscript: *“Dehydration reactions responsible for the blueschist to eclogite transition indeed releases a considerable amount of H₂O, which induces partial melting in the overlying mantle-wedge; such a process attests to the critical role of dehydration reactions in the seismicity as well as arc magmatism along subduction zones.” I don't see how dehydration reactions leading to mantle wedge partial melting attests to the role of seismicity.*

Reply: To clarify the influence of dehydration reactions on arc magmatism and seismicity, we have revised the statement with new references as below: “Dehydration reactions accompanied by the blueschist to eclogite transition indeed releases a considerable amount of H₂O. It has been suggested that this H₂O release can trigger intermediate-depth earthquakes via the dehydration embrittlement and induce partial melting in the overlying mantle wedge leading to arc magmatism.” (p3-4, line 74-76, reference #18, 23-25)

Comments made on the manuscript: *“The so-called ‘absence of blueschist’ is thus linked to the global dehydration and breakdown of the blueschist in the Precambrian plate tectonic settings where warm subduction was a predominant process for recycling of H₂O”. You should put this at the beginning of the sentence, before “The so-called”.*

Reply: We believe the present sentence deliver better the reasoning of *the so-called ‘absence of blueschist’* than placing *the Precambrian plate tectonic settings* in the beginning.

Comments made on the manuscript: “The Proterozoic thermal gradients of 25-50 °C/km in $\delta T/\delta \text{Depth}$ were based on the recent compilation of P-T data”. Inconsistent dimensions should be in $dT/d\text{Depth}$.

Reply: We have changed the unit on thermal gradient to $\delta T/\delta \text{Depth}$ throughout the manuscript (as marked in red color).

Comments made on the manuscript: Quartz?

Reply: We have specified the SiO₂ polymorph with corresponding mineral name, coesite in the text and the X-ray diffraction pattern in Fig. 2 (p5-6, line 123, 126, p11 line 272, and Fig. 2a).

Comments made on the manuscript: “In order to provide solid crystal structure data of the enhanced stability and a link to interpret seismic low-velocity layer along cold subduction zones, we determined the bulk modulus and linear compressibility of glaucophane at ambient and high temperature at 620 °C using various pressure media”. You present this data and don’t discuss it.

Reply: We have added descriptions on the potential impact of the derived anisotropic linear compressibilities of glaucophane on the seismic anomalies and the existence of low-velocity layers along the subduction zones (p6, line 134 and p9, line 216-221). We have also added references that link hydrous minerals to seismic anisotropy and delay times in subduction zones (Mainprice & Ildefonse (2009), Bezacier et al. (2010), Mookherjee & Bezacier (2012), Jung (2017), and Ha et al. (2019), reference #49-53).

Comments made on the manuscript: Multi-anvil press experiment (LVP) on glaucophane and Fourier-transform infrared spectroscopy (FT-IR) measurement would be moved to supplementary data.

Reply: These experiments were performed to verify the geotherm-dependent dehydration of glaucophane. As suggested, the multi-anvil synthesis and FT-IR measurement parts have been moved to supplementary information (Supplementary Fig. 6 caption, line 220-229).

Comments made on the manuscript: “the cold subducting slab with ca. 6 wt.% H₂O could remain hydrated towards the mantle transition zone.” Does it keep 6 wt% of water?

Reply: As mentioned in the introduction, fully hydrated oceanic crust with ca. 6 wt.% H₂O may release two-thirds and one-third of its fluids by 2 GPa along the warm and cold

subduction zones, respectively (p2-3, line 45-47, 53-55). The remaining water in the subducting slab would be transported to and released in the Earth's deeper interior. (p8, line 193-195)

Comments made on the manuscript: *“The low-velocity layers at the top of subducting slabs may reflect incomplete or continuous transformation from blueschist to eclogite extending to depths of 100-250 km”. It would be interesting to compile those seismic information based on cold vs warm subduction zone and show the relationships with glaucophane stability.*

Reply: We have addressed this issue as described in the key revision point #1 (Relationship between the stability of glaucophane and seismicity in the subduction zones). We now present the distribution of the earthquake occurrence along the representative cold and warm subduction system, i.e., the Tonga and Chile trenches, respectively, using data compiled at the International Seismological Centre (ISC) over the last 10 years. We have also added the references on the low-velocity layers and the earthquakes occurrences within (Abers (2005), reference #54 & Hacker et al. (2003), reference #58). According to these studies, it has been suggested that the generation of low-velocity layers and the occurrence of intermediate-depth earthquakes are related to the dehydration and the consequent release of volatiles during metamorphism (p9-10, line 224-227). In this regard, we confirm that the depth and frequencies of the earthquake occurrences show strong similarity with the stability ranges of glaucophane along the cold and warm subduction zones (revised Fig. 4 and new Supplementary Fig. 11).

Comments made on the manuscript: *“Dehydration embrittlement of serpentine was previously proposed as a possible mechanism for intermediate-depth earthquakes. Thus, as long as glaucophane is present in the subducting slab, its breakdown at depths may also induce dehydration embrittlement and provide another venue for nucleating intermediate-depth earthquakes.” Can you compare the stability of glaucophane with serpentine? Does it fit seismic data?*

Reply: As stated above, we now present the distribution of earthquake occurrence along the subduction zones to support that the depth and frequencies of the earthquake occurrences show strong coincidence with the stability ranges of glaucophane along the cold and warm subduction zones. In fact, glaucophane is a key mineral in the oceanic crust while serpentine is one of the principal hydrous phases in water-saturated peridotite (Schmidt and Poli (1998), reference #21). Although the stability field of serpentine has been estimated to extend down to ca. 150 km depth (Komabayashi et al. (2005) and Perrillat et al. (2005)), we intend to investigate the stability of serpentine in the comparative subduction conditions, i.e., water-saturated cold and water-depleted warm conditions, to deeper depths.

Comments made on the manuscript: *“Recent studies indicate that plate tectonics based*

on subduction-related processes has been initiated during the Proterozoic eon or as early as 3.8 or 4.4 Ga when the majority of subduction zones would be categorized as warm or intermediate-to-high thermal gradients system.” Episodic subduction yes, but not likely to be modern day like as numerous other have pointed out the role of blueschists in stabilizing the interface. With your study it would be interesting using curves of cooling Earth to estimate when the blueschists interface would be large enough to sustain modern like subduction

Reply: We appreciate such an insightful suggestion to link our results to the cooling curves of the Earth to estimate the onset of the blueschists interface. It is our intention to incorporate such multidimensional analyses in the future when extending the work as described above.

***Comments made on the manuscript:** “it may persist down to ca. 240 km depth under P-T conditions of cold subduction zone, facilitating the transportation of water into deeper Earth.” This is very interesting, but the paper should be rewritten in a more focus and for a broad audience.*

Reply: We appreciate such encouraging comments by reviewer #1. We are to maintain the balance, however, between describing our experimental results and reasoning their implications. We hope this revision would be both substantial and balanced enough to fit to the focused as well as broad readers interest.

***Comments made on the manuscript:** “Consequently, arc magmatism, volcanism, and related seismic activities linked to the dehydration of amphiboles have been globally suppressed, enabling blueschist to persist and be preserved in today’s geodynamic system. The ‘absence of blueschist’ from the Precambrian rocks might thus be explained by the subduction geotherm-dependent dehydration of glaucophane.” This needs quantification. What is the quantity of water lost in the process? Is that enough to suppress volcanic arc activity? Is this the only way to suppress arc activity?*

Reply: We present the estimations of the global H₂O influx in the revised Supplementary Table 4. Based on the average mass fractions of the rock forming minerals and their H₂O contents in the oceanic crust, the estimated H₂O influx from the dehydration of glaucophane would account for ca. 7-20 % of the total H₂O influx (0.4-1.2 wt.% H₂O) from the hydrated oceanic crust with initial 5-6 wt.% H₂O. We would therefore suggest that glaucophane could provide sufficient H₂O upon its breakdown to induce mantle melting. (p9, line 208-209, revised Supplementary Table 4). Although it is hard to affirm that the extended stability of glaucophane in the cold subduction environment is the only way to suppress the arc magmatism, we advocate that subducting glaucophane would play an important role in the global deep water cycle of the Earth (p11, line 254-255).

Comments made on the manuscript: Figure 2 is difficult to read and don't add much. It should placed in supplementary data.

Reply: We have revised Fig. 2 to enhance the comparison between different diffraction profiles as described in the key revision point #3.

Comments made on the manuscript: You seem to imply that cold subduction are more vertical than warm subduction in figure 4. This is not the case. The geotherm is mainly controlled by the convergence velocity.

Reply: As explained in the key revision point #2 and above, we have incorporated both the geometry and the temperature profile data into the respective cold (Tonga trench) and warm (Chile trench) subduction systems using the information from the ISC and the thermal models by Syracuse et al. (2010) (revised Fig. 4).

Reviewer #2 (Remarks to the Author)

Comments: This manuscript describes a series of high-pressure and high temperature experiments that seeks to chart the stability of the amphibole mineral glaucophane under the variety of thermal gradients of the Earth's crust. The authors have used a number of different experimental approaches to show that the glaucophane decomposing within the Earth's crust to albite and enstatite would occur areas of warm subduction, but not areas of cold subduction. The authors then point out the impact of this finding on release of water into the earth's subsurface over time.

Though the impact of the results is well reasoned, I have found it challenging to see how the experimental results have been reasoned. Much of this is because of the large range of experimental methods that are used and that the supplementary information does not give all the information require to ratify the results.

My chief concern with the manuscript itself is the presentation of the diffraction results, in Figure 2 and in the supplementary files, which though I can see demonstrate the general point that they authors claim, has not been presented to the rigorous standard that I would have expected. There are a number of concerns I wish to raise:

There is no presentation of the model fit to the data – only tick marks indicating the expected presence of Bragg peaks.

Reply: We appreciate the encouraging comments as well as constructive criticisms by reviewer #2. We have addressed the concern of the presentation of our diffraction data as

described in the key revision point #3 above. Through this revision, all our diffraction data have been analyzed and presented via profile fittings using the LeBail and Rietveld methods (revised Fig. 2, Supplementary Figs. 1, 3, 6, 8, 9, and new Supplementary Table 1), where all the diffraction peaks are marked with corresponding Miller indices.

Comments: I also assume the data have been reduced from their original format – especially the DAC measurements I would image that a background has been subtracted prior to the data being plotted in figure 2? Details of this should be given in the paper and it may help explain the differences in background between the PETRA and APS data.

Reply: As the reviewer pointed, our diffraction data contain different degrees of background components depending on the geometry of DAC, the X-ray energy used, types of detectors used, and the sample-to-detector settings, etc. We have added an explanation of the background subtraction in the revised Fig. 2 caption. We have also added the original XRD patterns before the background subtraction in the new Supplementary Fig. 3.

Comments: As the results of the paper depend on a series of results collected a different wavelengths it would be extremely beneficial for all the diffraction data to be plotted on a non-wavelength dependant x-axis (d-spacing or q space), currently it is really impossible (especially as the reduced data has only been made available in the figure) to compare each of them.

Reply: As the reviewer suggested, we now present all the diffraction data in the manuscript and supplementary files using the q-axis (\AA^{-1}) to better compare the peak positions in a wavelength-independent scale (revised Fig. 2 and Supplementary Fig. 1, 2, 3, 6, 8, 9).

Comments: By presenting the fits then the authors could also extract the individual scattering from each phase that contributes to the fit, and also explain why peaks that could be from a number of phases had been attributed to one (for instance the co-incidence of jadeite/enstatite peaks in Figure 4 a. I very much dislike the ‘labelling’ of the peaks that has been done, because it is not consistent (not all peaks are labelled) and not very informative – it could even be misleading.

Reply: As mentioned above, we have performed the profile fits on all our diffraction data to confirm the phase assignments using the LeBail method. We have marked the Miller indices for all the diffraction peaks in different colors to distinguish which phase contributes to which fit (revised Fig. 2 and Supplementary Fig. 2, 3, 6, 8, 9).

Comments: Is there a reason that a Rietveld refinement hasn't been undertaken on the single phase patterns of glaucophane at LT/LP conditions? Doing this for at least one of the

patterns (say the unheated pattern from the PE press) would show that there were not impurities and demonstrate the starting point of the experiments much better.

Reply: In this revision, we have performed the Rietveld refinement (Larson & Von Dreele (1986) and Toby (2001)) using the diffraction data of glaucophane measured at ambient conditions and present the results to confirm its phase purity and crystallographic details (revised Supplementary Fig.1 and new Supplementary Table 1).

Additional points that I would like to see addressed:

Comments: *There are no uncertainties quoted for the temperature measurements. This is especially important as the manuscript presents data where high temperatures were generated in a number of different ways. Though in some cases (the resistive heating for instance) an uncertainty in the measurement is described in the supporting information – this isn't the case for the sample subjected to laser heating. I would like to see an uncertainty quoted for every temperature measured in the manuscript.*

The same point can be applied to the pressure measurements too – again multiple pressure measurement methodology was used (ruby fluorescence, multi-anvil calibration and MgO diffraction) uncertainties of each measurement must be quoted in the manuscript for clarity.

Reply: We have quoted the uncertainties of the pressure-temperature measurements in the revised manuscript together with descriptions of the different methodology with corresponding references (“Methods” section in supplementary information, as marked in red color).

Comments: *Line 97 please add a reference to the Paris-Edinburgh press development*

Reply: We have added a reference for the Paris-Edinburgh press development (Kono et al. (2010), reference #32) (p4, line 100).

Comments: *Line 131, what is the relation of the compressibility parameters β_a , β_b , and β_c ? No equation (or reference to one) is given and how have these been derived from the collected data?*

Reply: We have added descriptions on the derivation of the linear compressibilities with a reference to the EOSFit program used in the pressure-volume/cell lengths fittings (revised Supplementary Fig. 4, reference #18). We have also added explanations on the potential impact of the derived anisotropic linear compressibilities of glaucophane on the seismic anomalies and the existence of low-velocity layers along the subduction zones. (p6, line 133-134 and p9, line

216-221)

Comments: Line 158 I wonder why that diffraction wasn't pursued for analysis of these samples?

Reply: We have performed the profile fitting on the diffraction pattern of the recovered sample to confirm the phase assignments. (revised Supplementary Fig. 6b).

Comments: Line 204 I'm concerned about the statement 'possible down to the upper mantle-transition zone (~410km)' – how is this justified by the experimental program as the pressures that the experiments went to were enough to replicate this depth?

Reply: In order to avoid any over-interpretation of our experimental results, we have removed the statement 'possible down to the upper mantle-transition zone (~410km)'.

Comments: Line 223 amend text to 'modelled by Nair and Chacko⁴⁴'

Reply: We amended the text to 'model by Nair and Chacko⁴⁸, (p9, line 213)

Comments: Figure 1, what is the Forbidden zone? This label is not referred to in the figure caption or in the manuscript text

Reply: We have added an explanation on the forbidden zone with a reference in the figure caption (Liou et al. (2000), reference #64) (Fig. 1 caption, line 543-545).

Comments: Figure 4, There is no indication of expected temperatures on this figure, could this be done with a colour scale?

Reply: As described in the key revision point #2 above, we have incorporated the geometry and temperature profile data into the respective cold (Tonga trench) and warm (Chile trench) subduction systems using the information available from the ISC and the thermal models by Syracuse et al. (2010) (revised Fig. 4).

Based on the above, we sincerely believe that we have addressed all the constructive criticisms and concerns of both reviewers to our maximum ability and therefore request our revised manuscript to be considered for publication in Nature Communications.

Please send any correspondence regarding this publication to:

Yongjae Lee

Professor
Department of Earth System Sciences
Yonsei University
Seoul, 03722
Korea

(office) +82-2-2123-5667 (fax) +82-2-2123-8169
(e-mail) yongjaelee@yonsei.ac.kr

REVIEWER COMMENTS

Reviewer #1 (Remarks to the Author):

Since the first submission the overall quality of the manuscript has slightly improved.

However, I still have two major comments:

1) In my opinion 3D for figure 4 is not useful and it should be turned into a larger, more readable and informative 2D figure to support the discussion,
On that point the authors answered that having a simple 2D thermo-mechanical model of subduction is out of the scope of their study. This is fine, but my suggestion was to have a consistent P-T framework from which they could explore the implications of their analytical study on the water fluxes in subduction zones using petrogenetic grids. For instance in Magni et al., 2014 (Deep water recycling through time), the figure 2 presents such framework. This allows to track to main metamorphic dehydration reactions and having similar presentation here would help broader the impact.

Without conducting 2D thermo-mechanical experiments the authors should use warm and cold subduction P-T fields (from analytical solution, check England and Wilkins, 2004, for instance) to illustrate the implications of their work on metamorphic reactions and water release, taking into account other hydrated minerals too.

2) The discussion is still partly unclear and repetitive, and should be carefully improved. For instance use geotherms to discuss the depth stability of glaucophane (instead of cold vs warm vs depth, which is rather unfocused). Having an improved fig. 4 would be of great benefit here. Another example concerns some information that are given but not really exploited e.g., L204-206.

I still do think this study brings important new constraints on water fluxes in subduction zone and I recommend it for publication after major revision.

Other comments are given in the pdf

Nicolas Riel

Reviewer #2 (Remarks to the Author):

Comments on the revision of 'The stability of subducted glaucophane with the Earth's secular cooling'

I am glad that the authors found my review constructive, and think that the presentation of the experimental results has vastly improved in this revision, to meet the standard I would expect for them to support their deductions. My major concerns have been answered, and the revision has improved the robustness of the study. I found the paper easier to read and believe that the study holds up to being of significant interest to the field.

Reviewer #3 (Remarks to the Author):

This manuscript by Bang et al. offers a fascinating new insight into the role of Glaucophane in

metamorphic dehydration reactions in subducting plates. In particular, the implications of Glaucophane dehydration for hypothesised changes in subduction zone seismicity and volatile cycle over time is fascinating. This result appears to corroborate the apparent absence of Precambrian blueschist in the rock record.

My expertise is in seismology and subduction zone tectonics, rather than metamorphism, so my few minor comments below relate to the "bigger-picture" implications of this result. Since these comments are very minor, it should not take too long for the authors to respond to these, and I very much look forward to seeing this compelling work published in Nature Communications soon.

1. Implications for volcanism and magmatism.

This work suggests that stability of Glaucophane at higher temperatures in the ancient Earth would have promoted more magmatism and volcanism, relative to current times. Are there any indications of greater volcanism / magmatism rates in the ancient (e.g. Precambrian) rock record? It would be good to have some further insights on this, if any exist.

2. Figure 4 and plotting of seismicity.

- I would like to see a depth axis with labels and ticks on Panels a) and b) of Figure 4.
- For the Tonga-Kermadec subduction zone (panel a), there are many earthquakes plotted that appear to show "flat-lying" lineations at certain depths (e.g. ~30km depth, 200 km, 300 km). These are default depths assigned to earthquakes where depth constraints are poor, e.g. due to lack of local seismic observations). So I would remove these default depth events from the plot to make it clearer. An alternative approach would be to plot events from the ISC-EHB bulletin, which is a subset of earthquakes with very high-quality depth constraints. This may make the image of subduction zone seismicity much sharper and easier to interpret.

3. Citing the International Seismological Centre (ISC) data.

Rather than just simply providing a link to the ISC webpage (reference no. 70), I think it is better to formally cite some of the papers relating to the ISC datasets with DOIs – see <http://www.isc.ac.uk/iscbulletin/citing.php> for more details. Or for the ISC-EHB datasets, these would be the papers to cite: <http://www.isc.ac.uk/isc-ehb/citing.php/>.

Review by Stephen Hicks, Imperial College London

Seoul, November 27, 2020

Dear Reviewers,

We appreciate all of the insightful comments and constructive suggestions for our manuscript to Nature Communications (NCOMMS-20-23298A) entitled: “The stability of subducted glaucophane with the Earth’s secular cooling” by *Yoonah Bang, Huijeong Hwang, Taehyun Kim, Hyunchae Cynn, Yong Park, Haemyeong Jung, Changyong Park, Dmitry Popov, Vitali Prakapenka, Lin Wang, Hanns-Peter Liermann, Tetsuo Irifune, Ho-Kwang Mao and Yongjae Lee*. Our point-by-point responses to all the comments and criticisms are summarized below (all the changes made in the revised version are marked in red).

Reviewer #1 (Remarks to the Author)

Since the first submission the overall quality of the manuscript has slightly improved. However, I still have two major comments:

Comments:

1) In my opinion 3D for figure 4 is not useful and it should be turned into a larger, more readable and informative 2D figure to support the discussion,

On that point the authors answered that having a simple 2D thermo-mechanical model of subduction is out of the scope of their study. This is fine, but my suggestion was to have a consistent P-T framework from which they could explore the implications of their analytical study on the water fluxes in subduction zones using petrogenetic grids. For instance in Magni et al., 2014 (Deep water recycling through time), the figure 2 presents such framework. This allows to track to main metamorphic dehydration reactions and having similar presentation here would help broader the impact.

Without conducting 2D thermo-mechanical experiments the authors should use warm and cold subduction P-T fields (from analytical solution, check England and Wilkins, 2004, for instance) to illustrate the implications of their work on metamorphic reactions and water release, taking into account other hydrated minerals too.

Reply: We have addressed the issue on Figure 4 by incorporating the established P-T framework/field and metamorphic dehydration reactions into our experimental results (with estimation on the H₂O flux). To outline the mineral phase assemblages of the subducting slab, we have used the model by Magni et al. (2014) for warm subduction and Hacker et al. (2003) for cold

subduction. To define the P-T field of the subducting slab, we found that the model suggested by Reviewer #1 (England and Wilkins, 2004) is focused on the temperature at the top of the slab and in the wedge. In order to provide a P-T field/framework of a subducting slab (with x-axis for distance from trench and y-axis for depth), we adopted two thermal models from Syracuse et al. (2010), i.e., slab dip, slab age, and convergence rate of 50 °, 110 Ma, 165 km/Ma for cold subduction and 45 °, 50 Ma, 60 km/Ma for warm subduction. We believe that the revised Fig. 4 with additional layers for such petrogenetic information for the corresponding cold and warm subduction slabs would allow the readers to compare the stability ranges of glaucophane to the established hydrous minerals assemblages and earthquake frequencies.

Comments:

2) *The discussion is still partly unclear and repetitive, and should be carefully improved. For instance use geotherms to discuss the depth stability of glaucophane (instead of cold vs warm vs depth, which is rather unfocused). Having an improved fig. 4 would be of great benefit here.*

Another example concerns some information that are given but not really exploited e.g., L204-206.

Reply: We have revised the manuscript to improve the clarity and to avoid the repetition. We have specified the geotherm conditions when we discuss the stability of glaucophane (p8-9, lines 177-178, 182-183) As stated above, we have revised Fig. 4 to present the models of glaucophane stability together with the observed seismic frequencies and established mineral assemblages in two contrasting geothermal gradient settings.

Comments: *I still do think this study brings important new constraints on water fluxes in subduction zone and I recommend it for publication after major revision.*

Reply: We appreciate very much the patience and continued efforts of Reviewer #1 to improve our manuscript for publication in Nature Communications.

Other comments are given in the pdf

Comments made on the manuscript: *“Over geological time, subduction systems with low thermal gradients such as the Tonga and Kermadec subduction zones have developed in the present global subduction environment as continued secular cooling of the mantle controlled the initiation of subduction-driven plate tectonics and steepening of subduction zones.” I don't understand this sentence. Rephrase.*

Reply: We have rephrased the sentence to “In turn, such a cooling in the average mantle temperature has affected the tectonic processes of the Earth by facilitating the modern-style

subduction with low thermal gradient slabs (~5-8 °C/km).” (p2, line 43-45).

Comments made on the manuscript: “but should also be rebalanced by the secular cooling of the mantle itself, manifesting the need to consider the geothermal gradients in the past and present tectonic settings to fully understand the evolution of deep water recycling.” I think you are talking about two different things here. Yes H₂O influx in deep mantle depends on subduction properties and hydrous mineral stability. The cause of having different subduction properties (PT, angle ...) is the secular cooling. Therefore the influx rate of water in the deep mantle changed as secular cooling occurs. You should rephrase to have the proper causality here.

Reply: We have rephrased the sentence to “The H₂O transport into the deep Earth is realized by subducting hydrous minerals, which exhibit a range of stability dictated by the P-T regime of the subduction system. It is therefore essential to investigate the stability of hydrous minerals as a function of diverse geothermal gradients in the past and present tectonic settings to fully understand the evolution of deep water cycling and related geochemical and geophysical activities.” (p2-3, line 49-53).

Comments made on the manuscript: “In general, it has been known that along the subduction zone, blueschist would be metamorphosed to eclogite via a suite of dehydration reactions involving the breakdown of amphiboles into pyroxenes and lawsonite into the garnet-kyanite-coesite assemblage at elevated P-T conditions.” Here you need to add those information on a modified figure 4. See main comment.

Reply: As stated above, we have revised Fig. 4 to add a new 2D layer for the known metamorphic assemblages as a function of the P-T fields in two contrasting geothermal gradient settings (revised Fig. 4).

Comments made on the manuscript: “ca. 240 km in certain cold subduction zones with lower thermal gradients.” give the gradient instead.

Reply: We have specified the value for the low thermal gradient of a cold subduction zone, i.e., ~5-8 °C/km based on the thermal models of Syracuse et al. (2010) (p8, line 177-178).

Comments made on the manuscript: “On the other hand, glaucophane decomposes into pyroxenes at shallower depth conditions of warm subduction zone, which is compatible with the blueschist to eclogite transition accompanied by the release of H₂O.” You need to show that on modified figure 4.

Reply: In revised Fig. 4, we have added a new 2D layer to compare the stability range of

glaucophane to the established mineral assemblages that represent the blueschist (BS) to eclogite (EC) metamorphism. We have also added the estimated amount of H₂O that can be transported through glaucophane, i.e., $0.7\text{-}2.1 \times 10^{19}$ kg H₂O in cold subduction zone (details are in the revised supplementary Table 4).

Comments made on the manuscript: *“along the geotherm of the warm subduction zone.” give values, if gradient is < x then glaucophane is stable until y depth*

Reply: We have specified the value of the geothermal gradient for a warm subduction zone to be ~8-12 °C/km and revised the sentence to “glaucophane decomposes into pyroxenes. i.e., transition to eclogite, at shallower depths between 50 and 100 km in warm subduction zones with a geothermal gradient of ~8-12 °C/km.” (p8, line 181-183).

Comments made on the manuscript: *The estimated water contents in the pre-eruptive magma are commonly in the range of 0-7 wt.%, based on previous studies using mineral/melt equilibria, melt inclusions in minerals, and stabilities of phenocryst assemblages” I don't see the need of this sentence here*

Reply: Our intention was to compare the water content in the magma to that in the oceanic crust, but we have deleted the above sentence as it is not directly related to our experimental results.

Comments made on the manuscript: *“According to our compressibility data” where are they available? reference?*

Reply: We have measured the bulk modulus and linear compressibility of glaucophane at ambient and high temperature, and the results are now summarized in Supplementary Fig. 4. We have added “(Supplementary Fig. 4)” after the above phrase” to refer to the compressibility data (p8, line 198).

Comments made on the manuscript: *“In a cold subduction zone of the northeastern Japan, seismic activities occur down to a depth of ca. 150 km.” Here you introduce an example but do not use it furthermore.*

Reply: We have revised the sentence to “Hydrous minerals have been suggested to be related to the seismic anisotropy and delayed seismic travel times along subduction zones in the depth range of 100-250 km. The observed anisotropy and stability of glaucophane could, therefore, account for such seismic anomalies distributed, which would be deeper in the colder and older slabs than in the warmer and younger slabs. Furthermore, seismic observations reveal that the low-velocity layers spatially coincide with the zones of intermediate-depth earthquakes, which is in

turn related to the dehydration of hydrous minerals.” (p8-9, line 199-205).

Comments made on the manuscript: “According to our study, glaucophane in the subducting oceanic crust persists down to P-T conditions of ca. 240 km depth along the geotherm of the cold subduction zone but only to P-T conditions between ca. 50 and 100 km depths (or near 60 km depth according to the blueschist rock and reactant mixture experiments) along the geotherm of the warm subduction zone.” You are repeating yourself quite a lot here.

Reply: We have revised the sentence to “With this regard, we show the correlation between the seismic frequencies along subduction zones and the stability range of glaucophane, i.e., the maximum depth of intraslab earthquakes ranges between 50-70 km in warm subduction zones whereas it extends down to over 200 km in cold subduction zones.” (p9, line 205-208).

Comments made on the manuscript: “thermal parameter showing average values of 48.9 °, 119.7 Ma, 74.8 km/Ma, and 6170 km, respectively” thermal parameter is 6170 km???

Reply: We have revised the sentence to “thermal parameter with average values of 48.9 °, 119.7 Ma, 74.8 km/Ma for slab dip, age, and convergence rate, respectively.” (p9, line 214-215). In addition, we have added the description of the thermal parameter with references as “The thermal parameter (ϕ) of a slab is defined as the product of the slab age (a), convergence rate (v_c), and the sin of the slab dip angle ($\sin(\alpha)$): $\phi = av_c \sin(\alpha)$ and is used to quantify the thermal state of a subducting slabs.” (in the caption of Supplementary Table 5). For example, the thermal parameter of Kermadec subduction zone is calculated to be $105.6 \text{ Ma} \times 64.6 \text{ km/Ma} \times \sin(56.1^\circ) = 5662 \text{ km}$.

Comments made on the manuscript: “Cold subduction zones then account for ca. 28.5 % of the global subduction system” Based on this, can you estimate the actual volume of water injected in the deep mantle? That would be interesting

Reply: We have estimated the volume of H₂O through glaucophane for cold subduction zones in the Supplementary Table 4 and marked it in the revised Fig. 4. It is estimated to be as much as 0.5-1.5 % of the whole ocean ($0.7\text{-}2.1 \times 10^{19}$ kg or ca. the volume of the Arctic Ocean). We also added the sentence as “The amount of H₂O transported by glaucophane in global cold subduction zones is estimated to be as much as ca. $0.7\text{-}2.1 \times 10^{19}$ kg, which is approximately the amount of water in the Arctic ocean.” in the discussion part (p8, line 193-195).

Comments made on the manuscript: “or the ratio of cold to warm subduction zone in the contemporary Earth is about 2 to 5” Delete the phrase.

Reply: We have deleted the phrase to avoid the repetition.

Comments made on the manuscript: “experimental results, glaucophane starts to dehydrate between 50 and 100 km range (or near 60 km depth according to the blueschist rock and reactant mixture experiments) under P-T conditions of warm subduction zone (or at a shallower depth near ~40 km in the Proterozoic tectonic setting with high thermal gradients). In contrast, glaucophane may persist down to ca. 240 km depth under P-T conditions of cold subduction zone, facilitating the transportation of water accounting for ca. 7-20 % of the total H₂O influx of the hydrated oceanic crust into deeper Earth.” You are repeating yourself again.

Reply: We have revised the sentence to “the dehydration depth of glaucophane has increased with decreasing thermal gradients hence with secular cooling, which would translate to transportation of water into deeper Earth (Supplementary Table 4).” (p9, line 221-223).

Reviewer #2 (Remarks to the Author):

Comments on the revision of 'The stability of subducted glaucophane with the Earth's secular cooling'

I am glad that the authors found my review constructive, and think that the presentation of the experimental results has vastly improved in this revision, to meet the standard I would expect for them to support their deductions. My major concerns have been answered, and the revision has improved the robustness of the study. I found the paper easier to read and believe that the study holds up to being of significant interest to the field.

Reply: We are happy to hear that our previous revision has met the standard of Reviewer #2 and appreciate very much the supports for the publication in Nature Communications.

Reviewer #3 (Remarks to the Author)

Comments: *This manuscript by Bang et al. offers a fascinating new insight into the role of Glaucophane in metamorphic dehydration reactions in subducting plates. In particular, the implications of Glaucophane dehydration for hypothesised changes in subduction zone seismicity and volatile cycle over time is fascinating. This result appears to corroborate the apparent absence of Precambrian blueschist in the rock record.*

My expertise is in seismology and subduction zone tectonics, rather than metamorphism,

so my few minor comments below relate to the “bigger-picture” implications of this result. Since these comments are very minor, it should not take too long for the authors to respond to these, and I very much look forward to seeing this compelling work published in Nature Communications soon.

Reply: We appreciate very much the encouraging comments and strong support of Reviewer #3 for publication of our manuscript in Nature Communications.

Comments: 1. *Implications for volcanism and magmatism.*

This work suggests that stability of Glaucophane at higher temperatures in the ancient Earth would have promoted more magmatism and volcanism, relative to current times. Are there any indications of greater volcanism / magmatism rates in the ancient (e.g. Precambrian) rock record? It would be good to have some further insights on this, if any exist.

Reply: As there are only a few identified regions of mega-caldera complexes and/or large igneous provinces dated for the Precambrian, it would be hard to estimate the degree of the volcanism/magmatism of the past compared to that of the contemporary Earth. We hope that such a modeling or investigation on the ancient volcanism would be available in the future.

Comments: 2. *Figure 4 and plotting of seismicity.*

*- I would like to see a depth axis with labels and ticks on Panels a) and b) of Figure 4.
- For the Tonga-Kermadec subduction zone (panel a), there are many earthquakes plotted that appear to show “flat-lying” lineations at certain depths (e.g. ~30km depth, 200 km, 300 km). These are default depths assigned to earthquakes where depth constraints are poor, e.g. due to lack of local seismic observations). So I would remove these default depth events from the plot to make it clearer. An alternative approach would be to plot events from the ISC-EHB bulletin, which is a subset of earthquakes with very high-quality depth constraints. This may make the image of subduction zone seismicity much sharper and easier to interpret.*

Reply: As suggested, we have added the depth axis in Fig. 4 and revised the 2D plot for the earthquake events using the data from the ISC-EHB bulletin with the high-quality depth constraints (revised Fig. 4 and Supplementary Fig. 11). As expected, the revised Fig. 4 shows clearer correlation between the frequencies of earthquake and the stability ranges of glaucophane along the cold and warm subduction zones.

Comments: 3. *Citing the International Seismological Centre (ISC) data.*

Rather than just simply providing a link to the ISC webpage (reference no. 70), I think it is better to formally cite some of the papers relating to the ISC datasets with DOIs – see <http://www.isc.ac.uk/iscbulletin/citing.php> for more details. Or for the ISC-EHB datasets, these would be the papers to cite: <http://www.isc.ac.uk/isc-ehb/citing.php/>.

Reply: As suggested, we have cited the papers related to the ISC-EHB datasets (in the caption of Fig.4, reference #66-68 and Supplementary Fig. 11, reference #24-26).

Based on the above, we sincerely believe that we have addressed all the criticisms and comments of the reviewers to our maximum ability and therefore request our 2nd revised manuscript to be considered for publication in Nature Communications.

Please send any correspondence regarding this publication to:

Yongjae Lee

Professor
Department of Earth System Sciences
Yonsei University
Seoul, 03722
Korea
(office) +82-2-2123-5667 (fax) +82-2-2123-8169
(e-mail) yongjaelee@yonsei.ac.kr

REVIEWERS' COMMENTS

Reviewer #1 (Remarks to the Author):

Since previous review all my major concerns have been answered. I still think that figure 5 should be in 2D but the manuscript is now in good shape and the results of major significance for the geosciences community.

Nicolas Riel

Dear Reviewer,

Seoul, January 15, 2021

Please find attached our 3rd revised version of the paper to Nature Communications (NCOMMS-20-23298) entitled: “The stability of subducted glaucophane with the Earth’s secular cooling” by Yoonah Bang, Huijeong Hwang, Taehyun Kim, Hyunchae Cynn, Yong Park, Haemyeong Jung, Changyong Park, Dmitry Popov, Vitali Prakapenka, Lin Wang, Hanns-Peter Liermann, Tetsuo Irifune, Ho-Kwang Mao and Yongjae Lee.

Reviewer #1 (Remarks to the Author)

Comments: *Since previous review all my major concerns have been answered.*

I still think that figure 5 should be in 2D but the manuscript is now in good shape and the results of major significance for the geosciences community.

Reply: We thank Reviewer #1 for his appreciation of our work and continued efforts to improve our manuscript. We are very pleased to hear from Reviewer #1 that our revised manuscript has addressed all the major concerns and is now in good shape for publication. Regarding Figure 4, we believe that the previously revised version has both 2D and 3D components to enhance the readability and hence advocate to use the current version.

Based on the above, we sincerely request our manuscript to be published in Nature Communications.

Please send any correspondence regarding this publication to:

Yongjae Lee

Professor
Department of Earth System Sciences
Yonsei University
Seoul, 03722
Korea
(office) +82-2-2123-5667 (fax) +82-2-2123-8169
(e-mail) yongjaelee@yonsei.ac.kr